# Magic of 5G Technology and Optimization Methods Applied to Biomedical Devices: A Survey

Lida Kouhalvandi [1], Ladislau Matekovits [2,3,4] and Ildiko Peter [5,*]

1 Department of Electrical and Electronics Engineering, Dogus University, Istanbul 34775, Turkey; lida.kouhalvandi@ieee.org
2 Department of Electronics and Telecommunications, Politecnico di Torino, 10129 Turin, Italy; ladislau.matekovits@polito.it
3 Department of Measurements and Optical Electronics, Politehnica University Timisoara, 300006 Timisoara, Romania
4 Istituto di Elettronica e di Ingegneria dell'Informazione e delle Telecomunicazioni, National Research Council, 10129 Turin, Italy
5 Department of Industrial Engineering and Management, George Emil Palade University of Medicine, Pharmacy, Science, and Technology of Targu Mures, 540139 Târgu Mureş, Romania
* Correspondence: ildiko.peter@umfst.ro

**Abstract:** Wireless networks have gained significant attention and importance in healthcare as various medical devices such as mobile devices, sensors, and remote monitoring equipment must be connected to communication networks. In order to provide advanced medical treatments to patients, high-performance technologies such as the emerging fifth generation/sixth generation (5G/6G) are required for transferring data to and from medical devices and in addition to their major components developed with improved optimization methods which are substantially needed and embedded in them. Providing intelligent system design is a challenging task in medical applications, as it affects the whole behaviors of medical devices. A critical review of the medical devices and the various optimization methods employed are presented in this paper, to pave the way for designers to develop an apparatus that is applicable in the healthcare industry under 5G technology and future 6G wireless networks.

**Keywords:** biomedical devices; fifth generation (5G); optimization methods; sixth generation (6G)

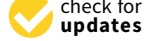



## 1. Introduction

Developments advancing the progress of wireless networking and big data technologies such as fifth and sixth generations (5G and 6G, respectively) are attracting more and more attention as they provide an opportunity to enhance transferring data [1]. These networks are preferable as they could deliver higher multi-Gbps peak data speeds and would support massive network capacity with ultra-low latency [2]. Recently, healthcare has become a hot research topic wherein medical treatments with the emerging 5G networks are combined together, leading to the high performance and precise treatment of disease [3]. In the current-day remedy procedure, various data have been collected from the human body through biomedical devices and sent to doctors through intelligent systems to analyze the diseases or to monitor the evolution of a recovery—all this in contrast to the conventional scenario, wherein doctors could make a decision with regard to determining the disease how the patient should be treated, as described in [4,5]. Biomedical devices can be on-body, off-body, and in-body depending on the medical application types where the in-body model includes implanted devices and on-body with off-body species are external devices for monitoring and controlling health issues [6,7]. In the field of body-centric wireless networks (BCWNs), antennas and amplifiers are two important components where antennas are either used for receiving or transmitting radio waves and amplifiers are typically employed for increasing the power of the signal. Hence, these essential devices

(i.e., antennas and amplifiers) must be properly designed as their output responses can affect the performance of the whole system [8].

Biomedical devices are significant tools as they can be used to diagnose, assist in delivering health care, and treat patients by providing access to the physiological data of patients [9]. These are employed for various targets such as medical imaging, bionics, and clinical engineering which use implanted devices [10]. Figure 1 illustrates one of the uses of biomedical devices as one of the active wearable sensors used for collecting the physiological data from a patient, before transferring the data to the hospital through the wireless networks. This task provides many benefits as it helps doctors diagnose critical diseases such as cancer in a shortest time with reduced cost, and paving the way for doctors to continuously consider the patient's body behavior [11]. Additionally, electroencephalogram (EEG), electrocardiogram (ECG), electromyography (EMG), and implanted devices are some of the other uses of biomedical devices through the 5G wireless networks [12].

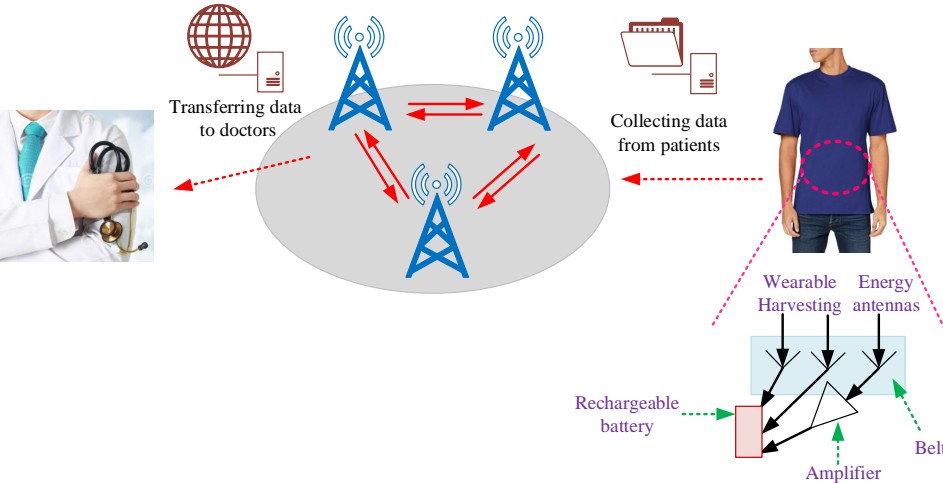

**Figure 1.** Conceptual diagram of a patient–physician relationship through wireless networks using active wearable sensors: here it includes antennas and amplifiers located in the belt of the patient.

Typically, there are two license-free distant operations of medical devices introduced by medical implant communication system (MICSs) and industrial, scientific and medical (ISM) bands, where the frequency range of MICS is 402–405 MHz and for the ISM band is 902–928 MHz, 2.4–2.5 GHz, and 5.725–5.875 GHz [13,14]. Biomedical devices, especially implanted medical devices, can be equipped with wireless systems, including (i) electronic circuits and (ii) antennas for exchanging and transferring data between various apparatuses [15,16]. Usually, the antennas can be of different structures such as dipole, loop, slot, and microstrip patch antenna; the diverse types of medical amplifiers comprise differential, operational, instrumentation, chopper, and isolation to gather and increase the signal integrity [17,18]. One example of the usefulness of devices such as smart physiological sensors is that the patient can monitor their health at a reduced cost.

This paper provides a comprehensive overview of the various biomedical devices employed in the recently published literature. Various used designs, simulation results, and challenges of the designs are highlighted and required challenges around the various techniques are provided. This research can pave the way for designers and engineers in more reliable and easily determining accurate and suitable designs/circuits. In many cases, this requires high-performance optimization methods. Some of them are also mentioned below.

This comprehensive review is organized into five sections. Section 2 presents the connection of biomedical devices with 5G and next-generation technologies. Section 3 describes in detail the various designed wearable devices in the recently published literature. Section 4 explains the used optimization methods in the field of biomedical devices. Section 5 presents the challenges. Finally, Section 6 concludes this manuscript.

## 2. Biomedical Devices through 5G and Next-Generation Technologies

Biomedical devices operating in the ISM frequency band are employed for investigating various physiological and biomedical signals [13]. The implantable and wearable devices make use of the wireless communication technology for safely receiving data from patients and transmitting any response data or enquiry from the doctors, in order to diagnose and monitor diseases in real time. Hence, in some cases, advanced and high-rate communication networks are required for transporting data. Figure 2 presents the overall view of the wireless communication network topology from a medical viewpoint as the wearable devices are gathering data from sick people and then transmitting data to the doctors for remote processing.

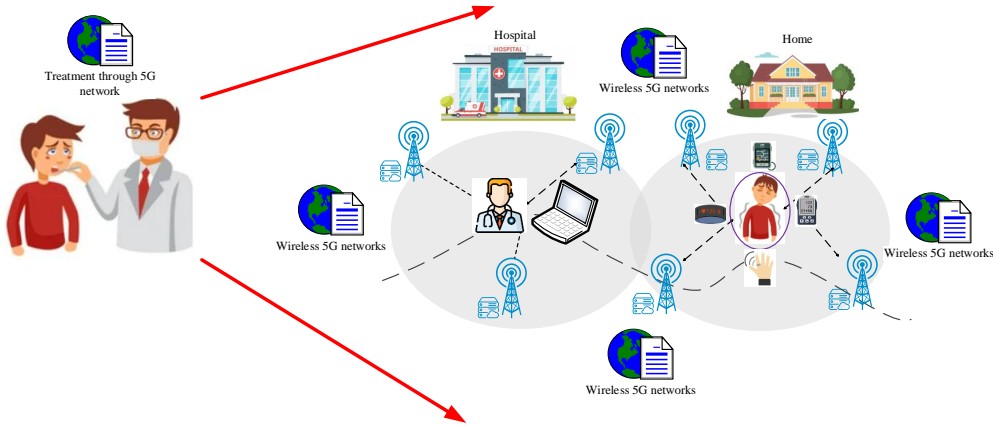

**Figure 2.** Use of 5G technology to remotely treating patients.

Following the success of 2G, 3G, and 4G technologies, 5G technology has been proposed for its higher throughput. This new technology aims to provide benefits to the development of Internet of Things (IoT) that incorporates wireless sensor networks, wireless mesh networks, wearable health care systems [19–21], etc. According to the report by the World Health Organization (WHO), a lack of healthcare workers is expected by 2035 [22]. Hereby, developing the self-health monitoring and intelligent wearable healthcare systems becomes very critical, also considering the aging population in the world.

The 5G networks provide potential useful possibilities for healthcare thanks to the unique characteristics in terms of speed, massive device connectivity, reliability, and so on [12,23]. These wireless networks pave the connection between the hospital and patients by commonly using integrated sensors and actuators [20].

The 5G technology, unlike Wi-Fi and Bluetooth, can handle the hazard of communication loss when connecting and managing the performances of several networks. Additionally, 5G networks can have suitable performance when there are multiple operators with various and many users that are expected to have different applications. Hence, 5G technology enables medical devices to provide reliable communication services with improved and critical life-supporting functions [23].

With the development of the 5G technology, 6G technology is expected to support even more mobile connections than 5G capacity and to provide data rates of 1 terabyte per second. These advantages can be very good candidates to the Internet of Things (IoT) which is slowly trying to be used in the medical applications that will be named as Internet of Medical Things (IoMT) [24].

Consequently, the importance of 5G technology from a medical perspective [12,25–28] can be summarized as follows:

- Improved speed of data transferring and large bandwidth;
- Advanced safety, health and security;
- Reduced waiting time (One millisecond end-to-end round-trip delay) for receiving data from wearable devices;
- Improved efficiency with capability to connect to many devices;
- Reduced power consumption;
- Advanced capacity for supporting devices;
- Possibility of the implementation of higher security standards.

### 3. Wearable Medical Devices

Wearable medical devices are essential tools for monitoring human's vital signs leading to improved healthcare. Figure 3 presents the typical sensing equipment used in medical devices.

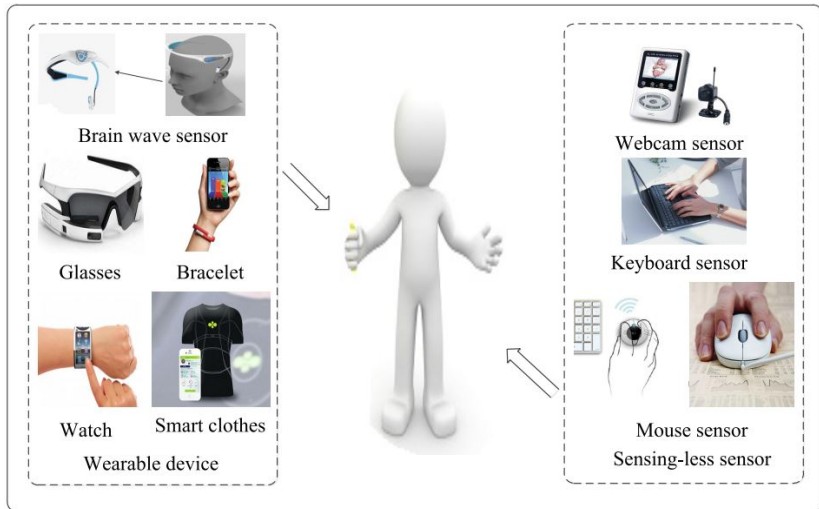

**Figure 3.** Various medical devices using sensors [29].

These types of devices can be either directly touching the skin or positioned/located on the clothes of patients. What is popular as a fabrication process is printing technology such as that of the flexible plastic foils attached to the body [30]. Wearable health technology products can be employed to monitor tears, temperature, wound, breath, sweat, and blood pressure [31–35].

Hao et al. presented the well-suited 5G wearable network able to provide low-latency data processing with appropriate resource utilization [36]. The presented network includes three parts as the 5G network, edge caching and computing, with network slicing (see Figure 4). Hereby, it supports the connection of a large amount of devices and utilizes the global cognitive engine for obtaining service-aware resource allocation.

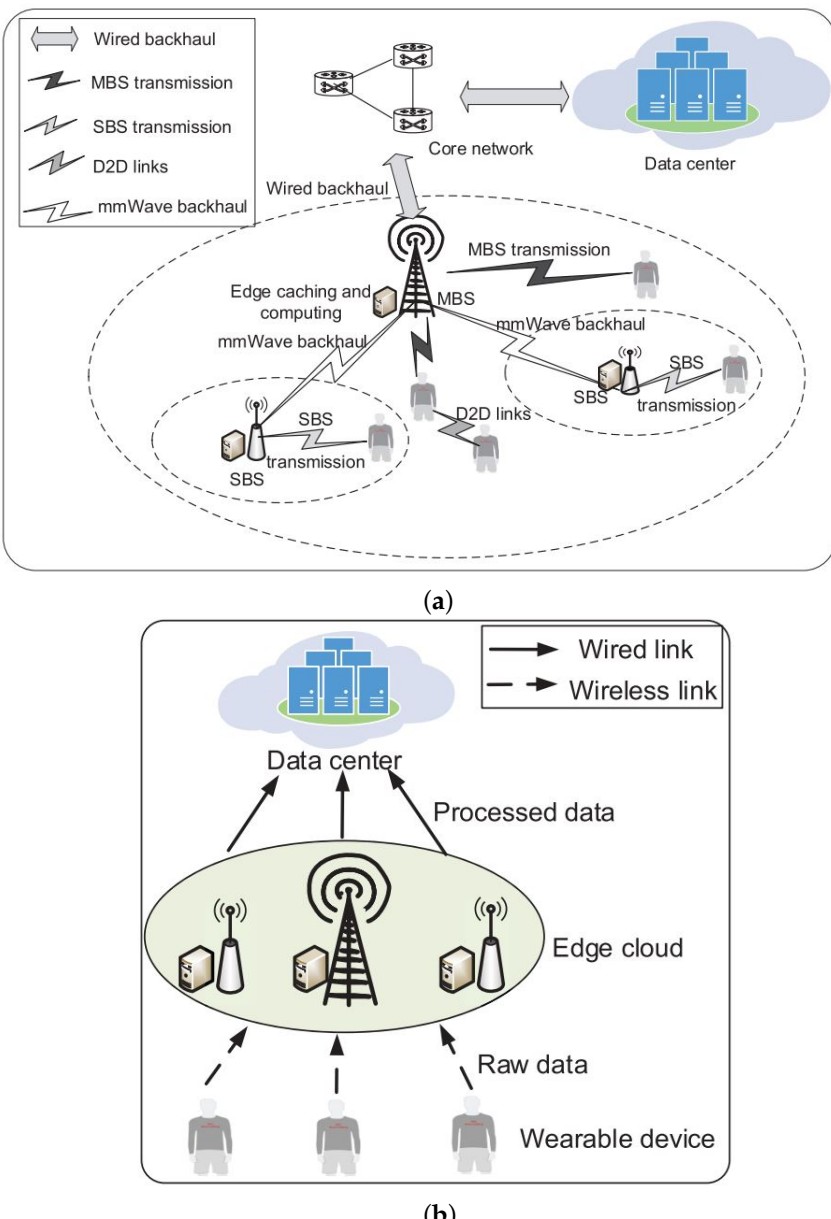

**Figure 4.** (**a**) The 5G ultra-dense cellular network in the wearable network macro-cell base station (MBS), small cell base station (SBS), device-to-device (D2D) (top); and (**b**) illustration of edge cloud and the data center (bottom) [36]. Reprinted/adapted with permission from Ref. [36]. Copyright 2022, IEEE.

Another aspect of using medical devices is to provide the 'self-management' of disease by using wireless local area networks (WLANs) and 5G wireless networks, for example, to control the long-term blood glucose by the patient to reduce diseases such as diabetes. In [37], an interactive telecare system (ITCS) is presented for diabetic sick people where IoT technology is employed for presenting the required action and alert service. The four important steps in relation to the proposed ITCS include: monitoring the patient; noting abnormal levels; notifying the caregiver and tracking the patient response; and finally noting any behavior changes are demonstrated in [37].

The Italian Ministry of Economic Development has developed a continuous home telemonitoring system that is appropriate for chronic patients and is presented in [38]. The system gathers the information and then the data are sent to the multi-edge computing server through 5G technology using environmental sensors.

Typically, these kind of devices use Internet of Things (IoT) to connect various apparatuses together through the Internet which increases the flexibility and compatibility of the overall performance. This connection can be provided using the smart sensors through wireless networks or Bluetooth [9,39].

The mm-wave range extends from 30 to 300 GHz; in some cases, the 20–30 GHz band is also considered where the WiGig protocol (IEEE 802.11ad) is appropriate for the wearable devices, such as reality glasses with video streaming capabilities [40].

The wearable technology which enables the medical industry to incorporate the data shared/achieved from the patient is provided to the doctors who can thus improving their patient's lifestyle. Wearable medical devices have various benefits including personalization, fast diagnosis, optimum, early decisions by doctors, and cost-effective remedy. In [41], a near-field magnetic communication system was reviewed which presents well-suited cochlear commercialized implantable devices that can save the lives of deaf people. Figure 5 presents the cochlear implant schematic that can be used in the ears.

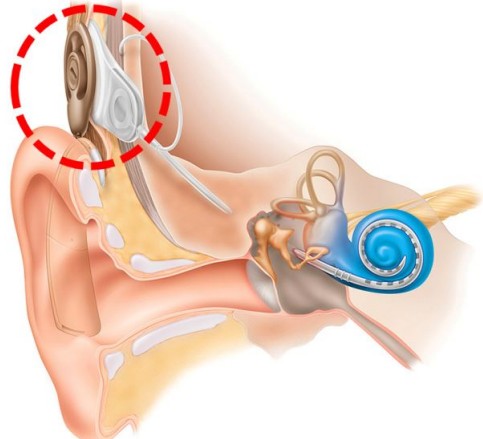

**Figure 5.** Illustration of cochlear implant (with a red dotted line) for a patient's ears [41].

In medical applications, what is so critical and important is gathering information from massive raw data. For this case, a new system structure is presented in [29] based on the emotion-aware healthcare which analyzes the relevance between emotion and sick people. Figure 6 presents the big data application in emotion-aware healthcare (BDEAH) where the 5G technology is employed into the structure and results in improved performance in whole system. As it can be noticed, the healthcare structure consists of a data center, software-defined network (SDN) components, 5G infrastructure, and sensing equipment.

The 5G technology is able to provide appropriate information infrastructure for emerging IoT applications. Guo proposed a medical dongle that is powered by the smartphones for monitoring and controlling the blood glucose and uric acid level [42]. Hence, by accessing these data, the medical doctors can provide accurate treatment and precise consultations.

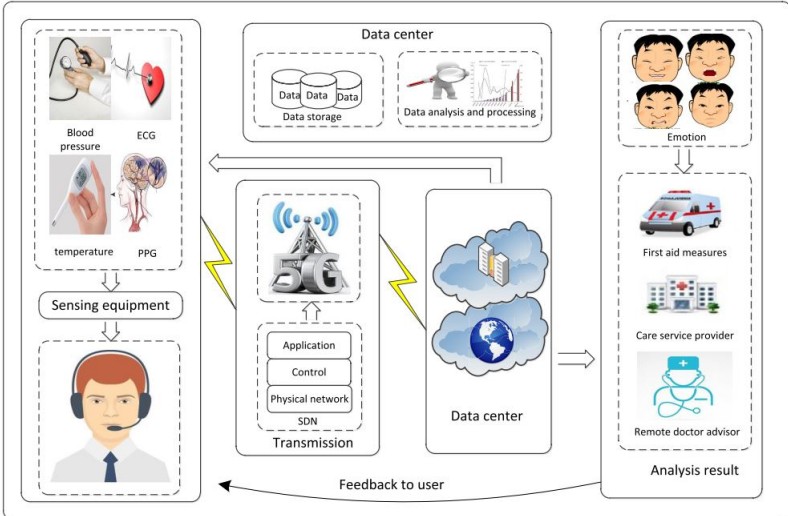

**Figure 6.** Presented emotion-aware healthcare in [29].

In [3], another magic aspect of 5G technology toward the diabetes is illustrated. In this manuscript, Chen et al. presented a 5G-smart diabetes system that involves smart clothing, smartphone, and big data clouds. The big data is employed for sensing and collecting information that the patients are suffering from. Figure 7 shows the data sharing for 5G-smart diabetes. In this personalized analysis model, firstly the 5G network, social network, and big data network are integrated for determining the interconnection. Then, after providing related data regarding the patient, by using the artificial intelligence methods, as machine learning, deep learning, and cognitive computing, the diabetes can be predicated in a smart and accurate way.

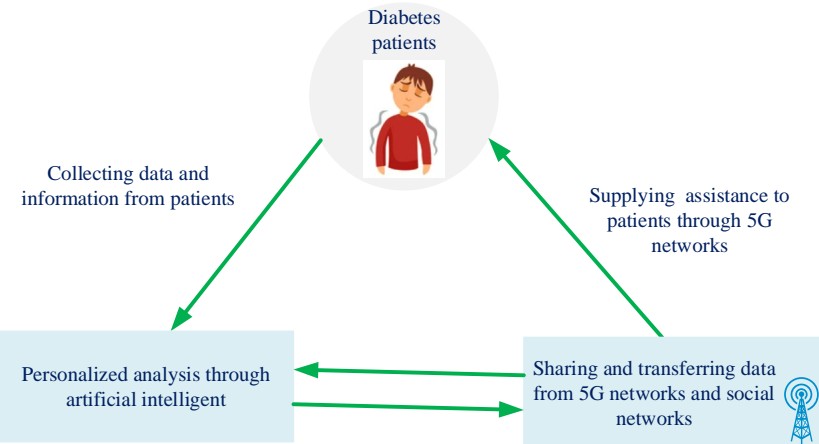

**Figure 7.** Data sharing procedure.

The use of wearable devices has increased during the COVID-19 pandemic wherein these tools helped improve healthcare quality by providing social and potential distance between hospital and patient [43,44].

Figure 8 presents the application of wearable devices during the COVID-19 pandemic to monitor the symptoms and to pave the way for doctors to control patients' healths over time [45]. L. Lonini et al. proposed a solution to detect the COVID-19 diseased state by collecting a number of physiological signals as heart activity, respiration, physical activity, and cough sounds using a body-conforming soft wearable sensor. As presented in Figure 9, by using this sensor placed on the suprasternal notch, the state of COVID-19 can be classified [46].

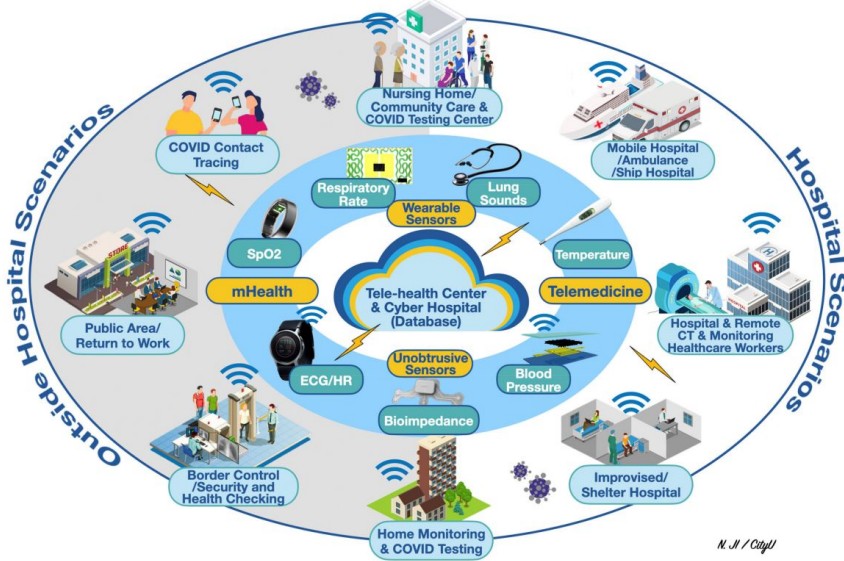

**Figure 8.** Practical use of medical wearable devices during the COVID-19 pandemic [45]. Reprinted/adapted with permission from Ref. [45]. Copyright 2022, IEEE.

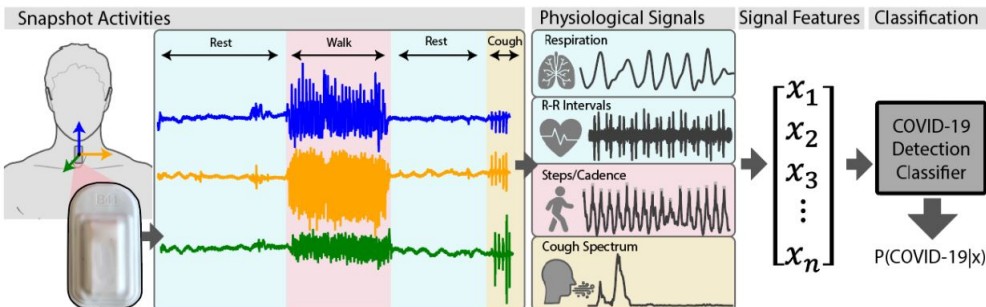

**Figure 9.** COVID-19 detection classifier using physiological signals [46].

In medical applications, one of the most important factors which must be considered is the security specification and the collected data from wearable devices are sensitive and require protection [47]. The protection security using the heart rate pattern is depicted in Figure 10.

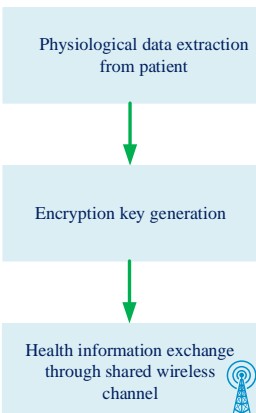

**Figure 10.** Intrawearable security solution.

L. L. Pon et al. presented a printed spiral resonator that is without external lumped elements and it is appropriate to be used in biomedical devices [48]. Figure 11 illustrates the fabricated printed spiral on the FR-4 substrate with the dielectric constant of 4.7 and

a size of 66 mm × 70 mm × 0.4 mm. The measured maximum power transfer efficiency (PTE) is 74.96% and PTE is more than 70% for all rotational angles from −180° until 180° at the central frequency of 13.56 (MHz).

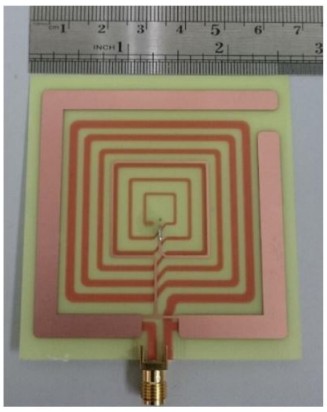 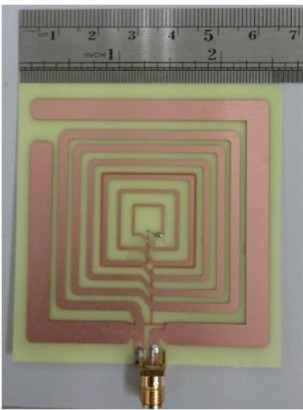

**Figure 11.** Printed spiral antenna proposed in [48]: top view (**left**) and bottom view (**right**).

N. Zhao et al. presented the procedure of characterizing the hand palm local channel at mm-Wave frequency bands of 27–28 GHz, 29–30 GHz, and 31–32 GHz, [49]. This study guides the handheld application developers and will develop low-power wireless sensors for medical and wearable devices. Figure 12 illustrates the hand palm local feature presented in [49], wherein two horn antennas and five human features are different in palm lengths. This design will help antenna designers obtain suitable efficiency in out of body-centric communication systems.

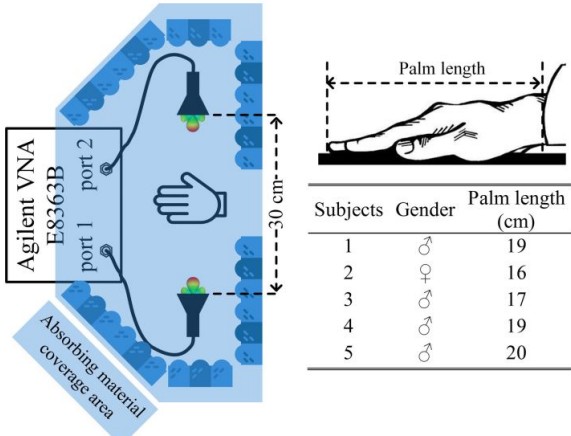

| Subjects | Gender | Palm length (cm) |
|----------|--------|------------------|
| 1 | ♂ | 19 |
| 2 | ♀ | 16 |
| 3 | ♂ | 17 |
| 4 | ♂ | 19 |
| 5 | ♂ | 20 |

**Figure 12.** The localized hand palm channel for the five test subjects [49].

In [50], a dual-band dual-polarized wearable button antenna array is presented that covers the two frequency bands of 4.50–4.61 GHz and 5.04–5.50 GHz which are suitable for 5G applications (sub-6 GHz band). This antenna array is based on the crossed dipole topology and is analyzed in free space and on the body. Figure 13 shows the antenna structure in detail where it can replace a button on clothing and thus be wearable.

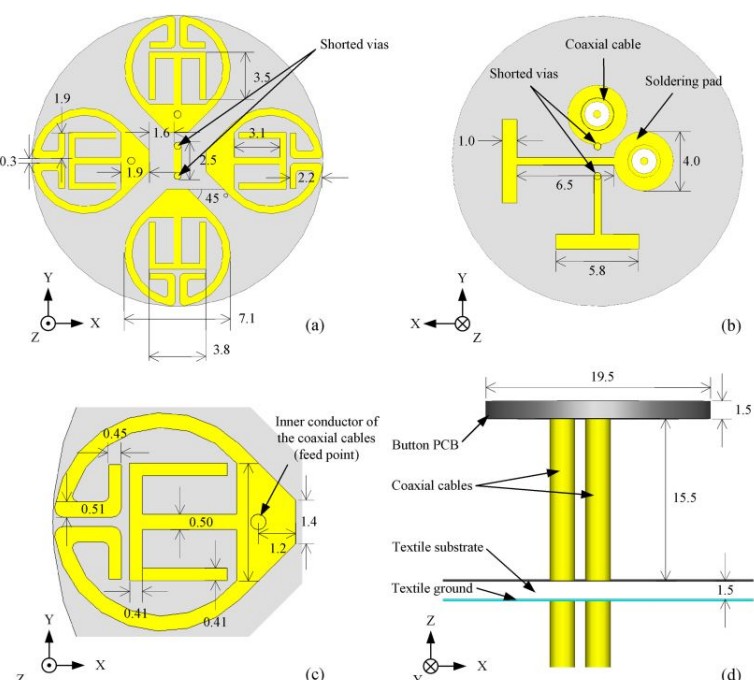

**Figure 13.** Illustration of the wearable button antenna. Reprinted/adapted with permission from Ref. [50]: Copyright 2022, IEEE. (**a**) top view; (**b**) bottom view; (**c**) radiator structure; and (**d**) side view.

From research and industry perspectives, monitoring sleep has become important in the sleep healthcare systems. In [51], a new sleep healthcare system based on radio frequency science for leveraging signal processing was presented, namely Sleep Guardian. This system can detect and notify of any abnormal situation arising when a patient is sleeping; additionally, it can be integrated in present Wi-Fi infrastructure. The healthcare systems are connected through the base stations and the required data are collected through accumulated healthcare big data where machine learning technology is employed for predicting and better diagnosing patients.

Carneiro et al. [52] proposed electroencephalography (EEG) biosignals' acquisition—that is reusable—on e-textile and convenient. The proposed fabric-based headband tackles the drawbacks of the bulkiness of the system, an unsuitable situation caused by the rigid electrodes, and laborious tasks. In this study, the proposed headband where the printed electrodes are getting advantage of a very low skin-electrode impedance.

Occhiuzzi et al. presented a dual chip flexible epidermal antenna that is for measuring the skin parameters using the UHF Radio Frequency Identification (RFID) technology inside the frequency band of 860 MHz–960 MHz [53]. Figure 14 describes the use of this proposed RFID that is used for monitoring health by directly sampling from the skin of patients. The used antenna is with two independent broadside radiating modes and the results present the turn-on powers ($P_{to}$ (dBm)) with the differential power ($\Delta P_{to}$) that the chip is attached to different parts of the body such as the chest, forehead, abdomen, and hand, where the antenna is within a distance of 50 cm from the body.

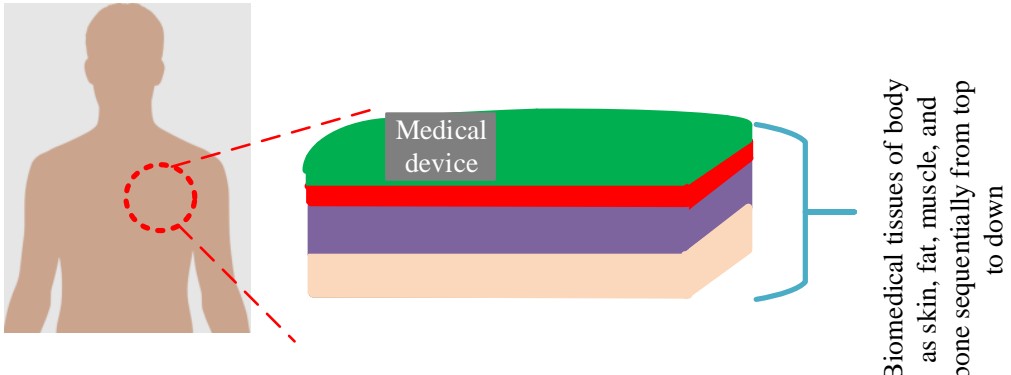

**Figure 14.** RFID that is used on the skin of humans.

In [54], a new flow-guided nano-communication network was presented that can be used in up-to-date medical applications to monitor, collect, and transmit various data from the human body (Figure 15). This network can be operated in the bold circulatory system. The presented optimization in this paper leads to maximize throughput by determining a suitable frame size as a function of the available energy, nano-router location, and transmission rate.

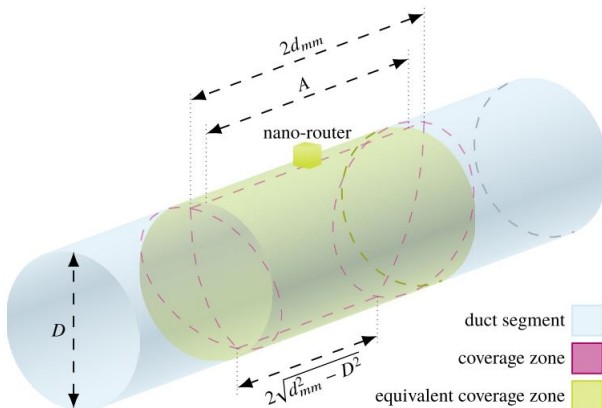

**Figure 15.** Illustration of flow-guided nano-network [54].

## 4. Optimizations for Healthcare and Biomedical Devices

Optimization term refers to achieving the best trade-offs between the targets and constraints (see Figure 16) and improved optimization methods are required as the designs become complex [55,56]. For this case, firstly the design variables are defined for each of the biomedical devices, then the suitable optimization method is selected, aiming to optimize the targeted output specifications. Due to the requirement of necessity, the designer selects the output specifications and then determines the suitable optimization method for achieving the desired outputs. For example, medical designers may need to enhance the bandwidth or gain of biomedical devices. Hereby, an appropriate optimization method for their problem will be selected that aims to improve the bandwidth or gain.

Importantly, in the field of healthcare, advanced technology and methods are required to provide and diagnose whether negative feedbacks exist in live bodies to alert and save lives in a timely manner [57–68]. Hereby, this section provides a comprehensive overview of the various employed optimization methods on the biomedical devices.

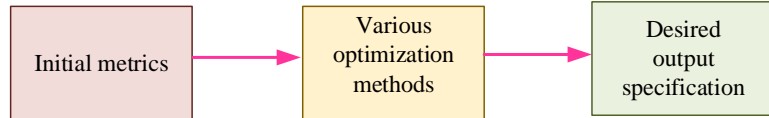

**Figure 16.** Aim of the optimization process.

### 4.1. Optimizations for the Mobile Cloud Computing (MCC) Technique

Mobile devices are another important tool that can provide healthcare services under any condition and time (see Figure 17). The mobile cloud computing (MCC) has been recognized as an effective method for healthcare services. For developing the MCC technique, various reported optimization methods are summarized by X. Wang and Z. Jin in [69].

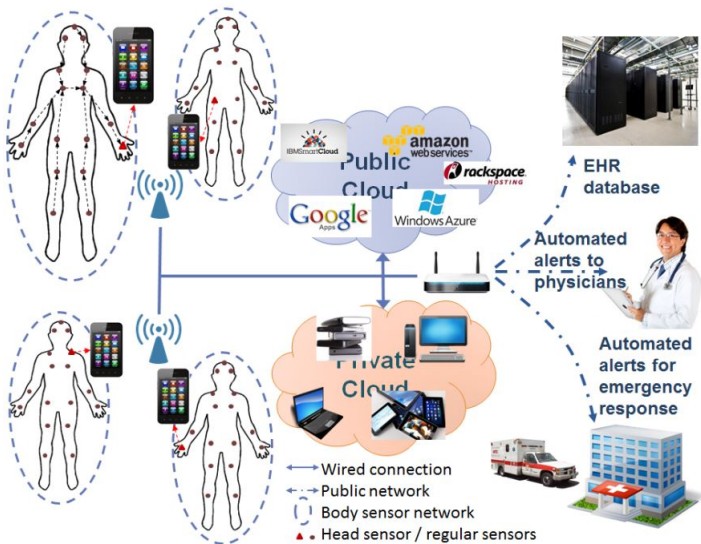

**Figure 17.** Perspective representation of the MCC technique used in healthcare [69].

### 4.2. Geometry Optimization

For implantable medical devices, wireless energy transferring is developing exponentially and the most famous solution is that of inductive powering. The inductive coils misalignment is the main drawback in wider wireless powering and is solved by the geometry optimization presented in [70]. A new algorithm for the coil couple design is presented where it provides a performance metric where the maximum and minimum targeted values of the load power are defined. This method ensures the stability and suitable output specifications, and it is formal procedure that can be employed in the computer-aided design (CAD) tool. The presented geometry optimization can optimize the transmitting coil external radius, transmitting coil turns number, coils internal radii, and receiving the coil turns number.

In [71], geometrical coil optimization was performed, which is employed for optimizing planar spiral coils with integrated capacitive elements for enhancing the power transfer efficiency while reducing the size of transmitter and receiver configurations. The proposed method results in a maximum efficiency of 31% with the existence of biological tissues where a 30 mm primary and a 10 mm secondary coil is sized (see Figure 18).

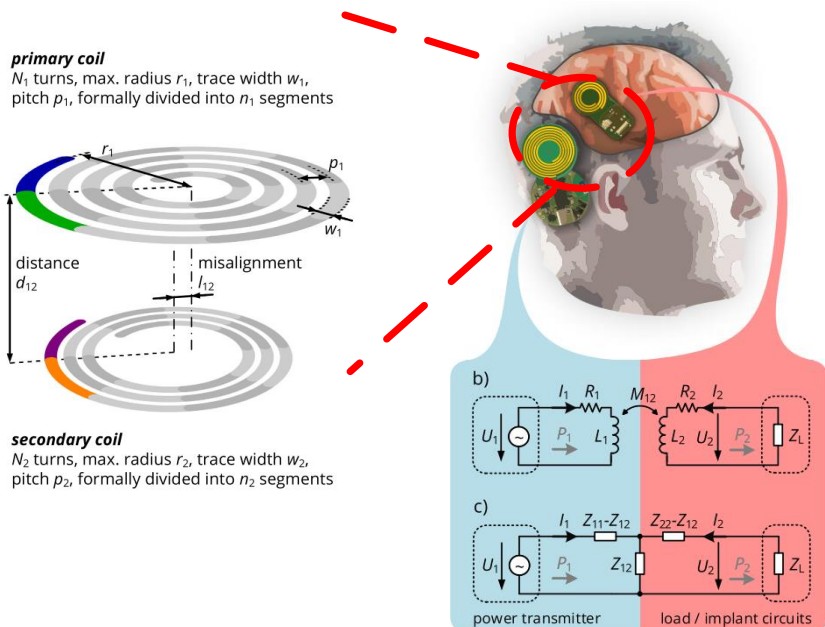

**Figure 18.** Geometrical coil optimization used for brain neural implant [71].

### 4.3. Optimizations Using the Artificial Neural Network (ANN)

Recently, artificial intelligence, big data analytic, and machine learning have become successful optimization tools and become subjects of industrial interests [72–74]. Zheng et al. presented the use of an artificial neural network (ANN) for estimating the magnetic resonance imaging (MRI) radio frequency (RF) exposure for the implantable plate system where it considers the existing problems using the non-linear and high-dimensional features [75]. The general performance of the ANN is improved by using the mean impact value (MIV) and genetic algorithm (GA). The presented ANN in [75] includes six inputs, two hidden layers, and one output layer with the transfer function of 'tan-sigmoid' where the last outcomes are considered using the mean square error (MSE) algorithm. In this manuscript, the MIV algorithm is employed for deciding the inputs of the ANN and feature selections; additionally, the GA method is for optimizing and adjusting the weights and bias of the ANN.

In [76], deep neural network (DNN) architecture is presented to be used in the biomedical image segmentation. The proposed method is named the evolutionary compression DNN method (ECDNN), where during the evolution, this method can simultaneously optimize the network loss and the number of parameters. In this study, the framework of the proposed ECDNN that is useful for biomedical image segmentation is presented. It consists of three steps where firstly the convolutional neural network (CNN) is constructed using the training dataset. Then, ECDNN reduces the size based on the CNNs and the searching of Pareto-optimal optimization. Thirdly, the final solution is selected from the Pareto-optimal set to compress the original model and update the method if required. The presented procedure is fast as it finds a solution for compressing CNNs in a single run.

In clinical uses, the accuracy and robustness factors of brain–machine interfaces (BMIs) are key important specifications as BMIs could provide a situation for driving the robotic arm and computer cursor. Liang and Kao proposed the DNN method for optimizing encoders and synthesizing the motor cortical neural population activity [77] (see Figure 19).

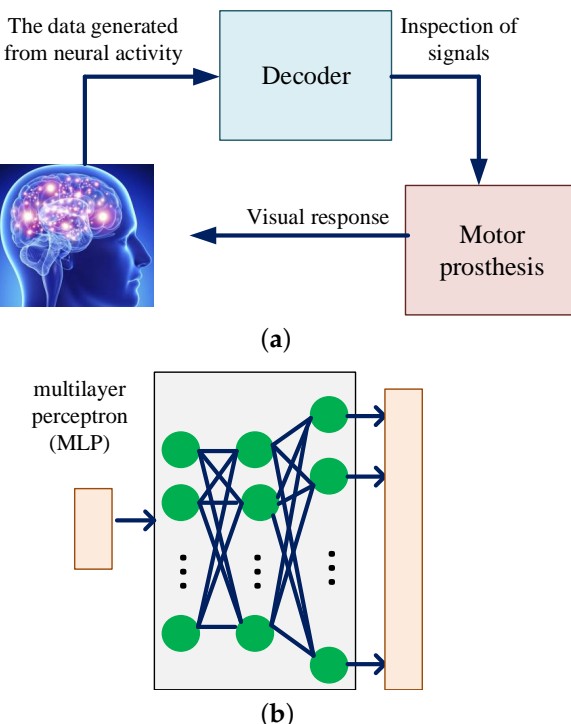

**Figure 19.** (**a**) BMI system and neural encoder (**top**); (**b**) multilayer perceptron (MLP) architecture (**bottom**).

In [78], the neural network (NN) is employed for use in electrocardiogram (ECG) processor. For accurate authentication, the front-end signal processing of ECG signals and back-end NNs is presented. The NN is used for minimizing and maximizing the intra-individual distance and inter-individual distance over time, respectively. Figure 20 presents the schematic of the ECG processor that includes the front–end signal processing modules and back-end NN-based ECG feature extraction modules. The proposed system achieves 62.37 µW power consumption at 1.2 V supply voltage with a low equal error rate (EER) of 1.36%.

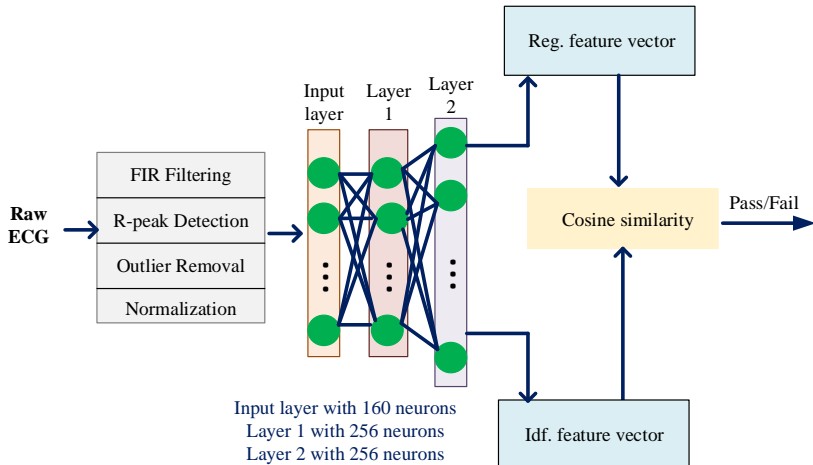

**Figure 20.** Employed NN for ECG processor.

Image segmentation and classification is another promising computation in diagnosing diseases. R. J. S. Raj et al. presented the classification DNN for diagnosing lung cancer, brain image, and Alzheimer's diseases [79]. The opposition-based crow search (OCS) algorithm was employed to enhance the performance of the DNN, leading an optimal feature selection being derived. Figure 21 illustrates the flowchart of the presented classification DNN

resulting in an accuracy of 95.22%, a sensitivity of 86.45%, and a specificity of 100% for the used medical images.

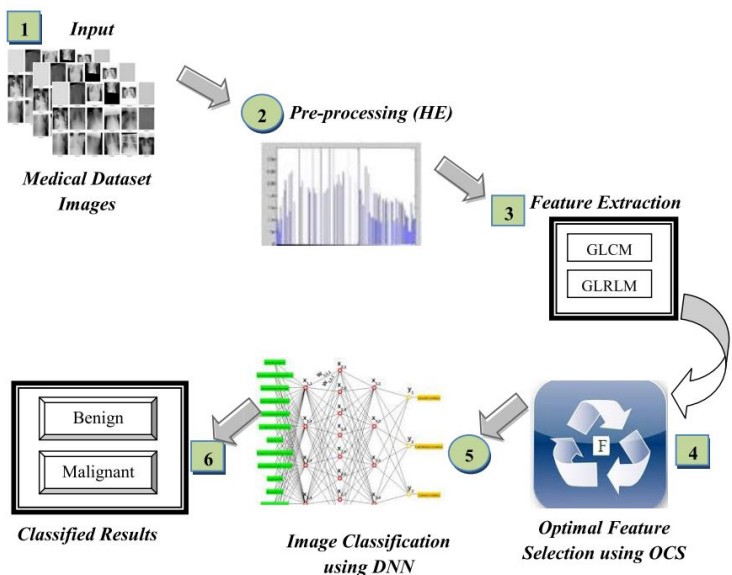

**Figure 21.** The presented classification DNN with the use of the OCS algorithm in [79].

S. S. Sarmah presented the IoT-centered deep learning modified neural network (DLMNN) that is performed in three steps as follows: authentication, encryption, and classification [80]. The employed classification DNN determines whether the patient is suffering from heart disease and in the case of an abnormal situation, the system alerts the physician. Figure 22 summarizes the proposed methodology wherein patients must connect with IoT and the presented DLMNN collects and performs an analysis of the extracted data with the security of 95.87%.

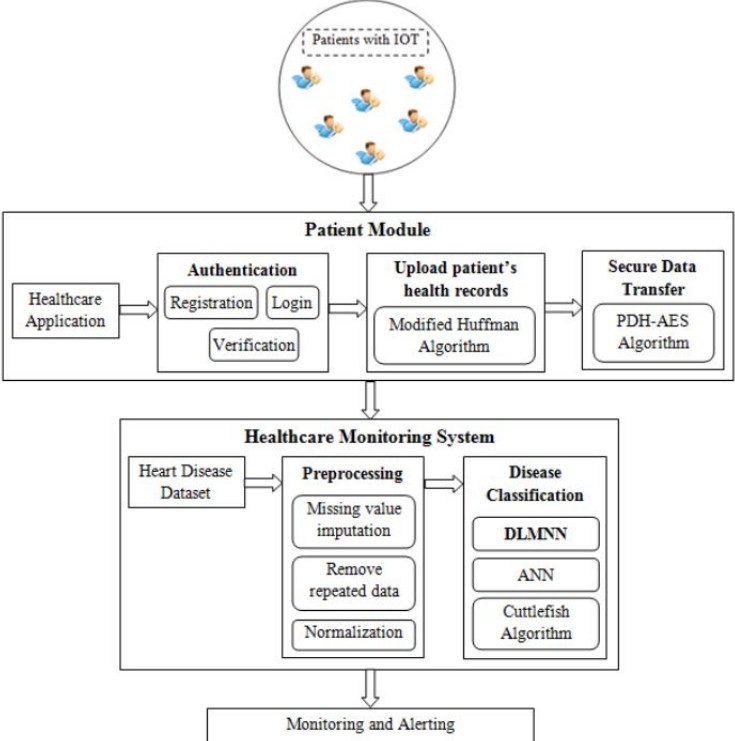

**Figure 22.** The presented classification DNN with the use of the OCS algorithm in [80].

### 4.4. Dimension Reduction Method

Tele-ophthalmology is an efficient method of diagnosing diseases during the early stages by extracting optical images from parts of body. In [81], RetinaMatch was presented, that developed a promising retinal matching technique with the combination of dimension reduction and mutual information (MI). The typical idea in the provided matching technique is to narrow the optimization domain as a coarse localization process and the patient's image are mapped and compared with the RetinaMatch, leading to an accurate diagnosis. Figure 23 presents the employed method using the coarse localization process with the descriptions around the principal component analysis (PCA).

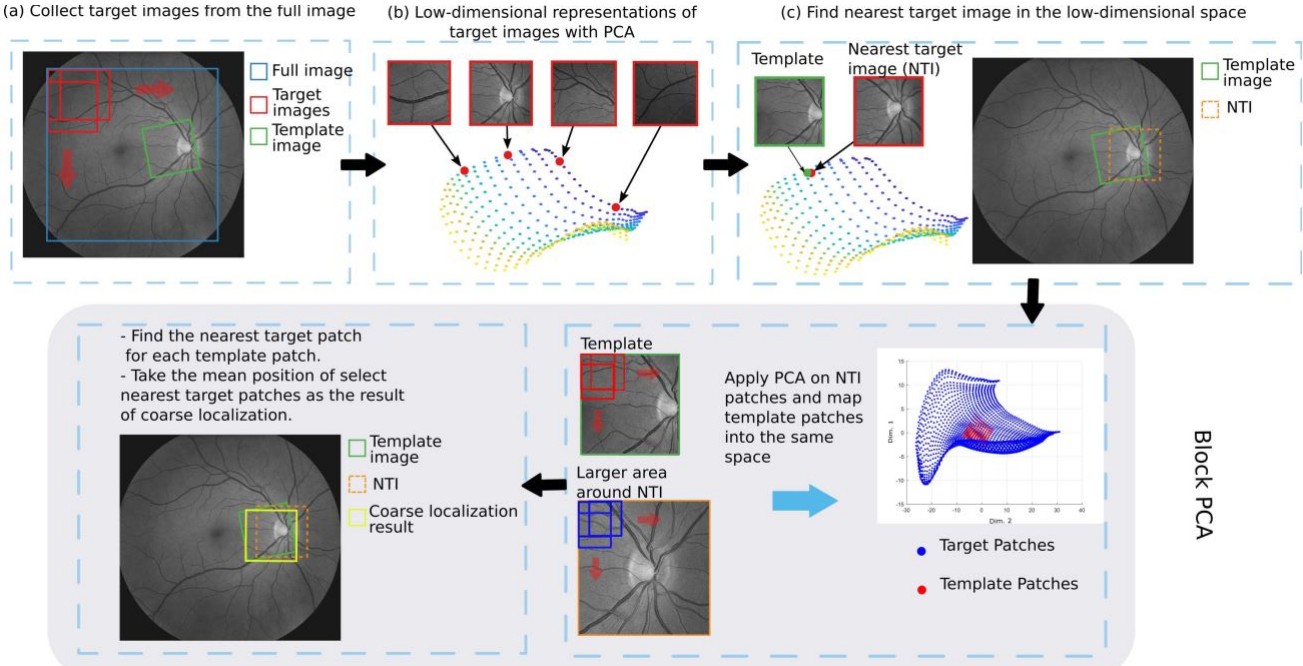

**Figure 23.** Details of RetinaMatch presented in [81].

### 4.5. Practical Gait Feedback Method

Meng et al. proposed a practical gait feedback method that requires suitable feedback [82]. This method does not need an alignment of the inertial measurement unit (IMU) and consists of a hierarchical model (i.e., two-layer model). In the gait measurement system, the high-level layer is a Bayesian recognition algorithm employed for observing the gait phases and in the low-level layer, the calculation of the measured ankle plantar/dorsiflexion angle from acceleration and angular rate data is completed. This construction leads to enhance the efficiency of gait interventions.

### 4.6. Stochastic Optimization Algorithm

In [83], a method for exploring physiological conditions based on an integrated stochastic optimization algorithm is presented (see Figure 24). The proposed framework consists of levels as the generation, evaluation, and execution where the input is the clinical knowledge of the heart model parameters. This automated construction safely and efficiently help manage heart failure under various physiological conditions.

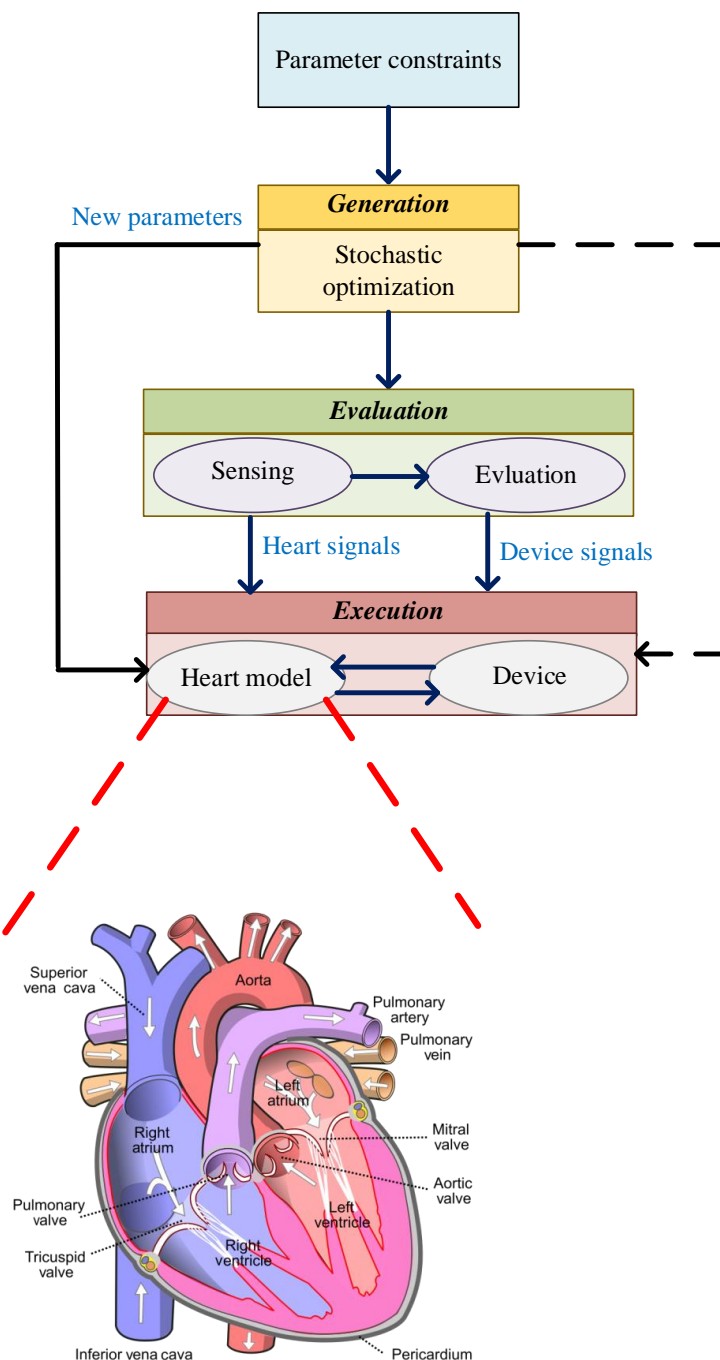

**Figure 24.** Closed-loop device based on the stochastic method with the illustration of a cardiac conduction network where circles shows the clusters of tissues.

### 4.7. Swarm Optimization

In the framework of IoMT, a healthcare monitoring system based on machine learning techniques and modified salp swarm optimization (MSSO) algorithm is presented in [84]. It is employed for predicting blood pressure (BP), age, sex, chest pain, cholesterol, blood sugar, etc., which are risks of heart disease. Figure 25 presents the health-monitoring system with the employment of a Levy-based crow search algorithm (LCSA), modified salp swarm optimization (MSSO), and an adaptive neuro-fuzzy inference system (ANFIS) processes.

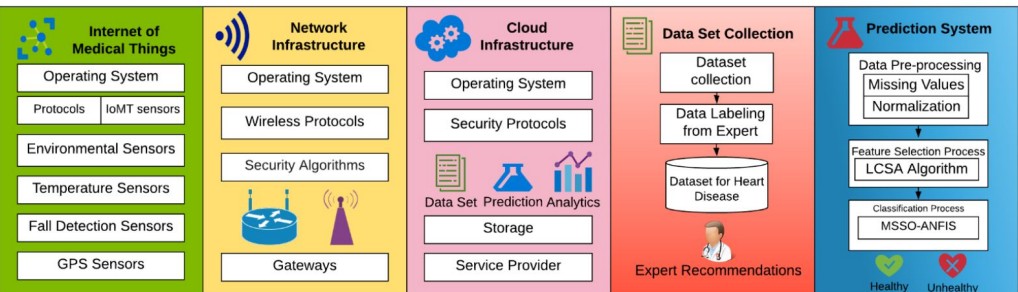

**Figure 25.** Illustration of the presented health monitoring for detecting heart diseases [84].

*4.8. Nested Optimization*

For the use of custom foot orthoses (CFOs), in [85], rapid evaluate and adjust device (READ) prescription methodology was presented with the combination of follow-up and design feedback loops. This methodology leads to fast evaluation with improved accuracy. In this manuscript, the 3D ergonomic measurement system (3DEMS) is presented where the 3D database is firstly provided and then using the scanned plantar surface, the arch is partitioned. The nested optimization method is able to provide the optimal representative segments.

## 5. Challenges and Future Trends

As discussed in the previous sections, biomedical devices have growing exponentially over the last ten years and have influenced the industry. Due to some challenges, as in other fields, the biomedical industry is facing disruptions. Table 1 presents in summary the various high-performance biomedical devices with use in different applications. Hereby, this section is devoted to introducing some of the important issues for biomedical devices from the practical industry point of view.

*5.1. Security Concerns*

Medical devices are growing and developing day-by-day and it is not essential to be settled in clinical places. They can be placed at homes near patients—using both wired and wireless channels—to continuously care for and record the data of sick people. This state-of-the-art situation created by the biomedical and wearable devices tremendously reduces the cost of healthcare. Hereby, one of the factors that has become important is that of security. This matter is significant as the achieved data from patients are subject to legal constraints; also, the lack of privacy may physically harm the patients due to the modified or suppressed data [86–88].

For the medical devices that are working and communicating through the wireless networks, interruptions from unauthorized people can generate problems and deep hazards. In simple words, any remote offense can provide incorrect and false data that result in inaccurate treatments which can be harmful [89].

The 'endpoint Protection' using the antivirus 'cybeats' where cryptography and vulnerability analysis are employed [90–92]. The cryptography is the science of secure communications techniques where just the sender and intended recipient can obtain the message, as presented in Figure 26.

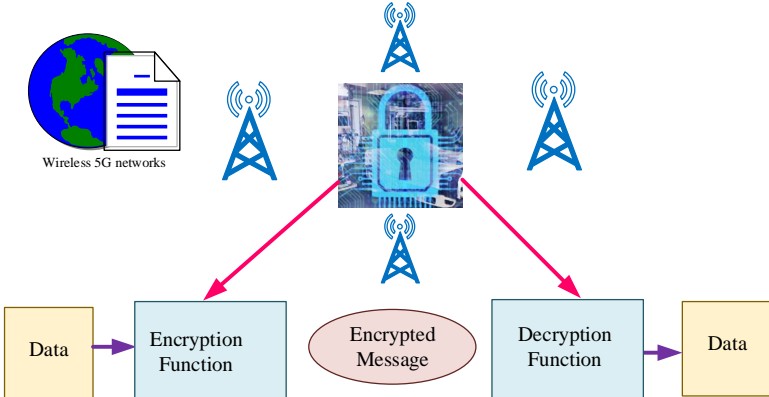

**Figure 26.** Cryptography science as a solution for providing security in biomedical devices.

To any system, security is a significant issue and the medical devices typically comprise of heterogeneous elements. Nowadays, medical data are transferred by wireless networks due to the high reliability and speed. For this case, effective and trustworthy components and systems must be employed for taking care of patients' physical security and data privacy, as well as of the physical security and data security of medical devices. For 5G wireless networks, cryptography science can help in transmitting data from patient people to doctors in an effective and healthy way.

**Table 1.** Summary of some reported biomedical studies with the focuses and output specifications of each one.

| Ref. | Scope | Contribution | Specifications |
|------|-------|--------------|----------------|
| [42] | Provides concerns related to the future wearable systems. | Introduces the system level dynamics that governs next-generation personal networks. | Meeting the requirements of WirelessHD, and ECMA-387 standards. |
| [29] | Constructing a system for joining the emotion computing and emerging communication technologies. | Uses the 5G technology for accelerating the original data collection. | Includes emotion and cloud computing with software-defined network. |
| [37] | 5G healthcare with medical Internet of Things (MIoTs). | Presents a medical dongle for blood glucose and uric acid monitoring. | Uploading the medical images via online medical platform. |
| [3] | 5G-Smart Diabetes. | A system including smart clothing, smartphone, and big data clouds. | Personalized diagnosis and treatment suggestions. |
| [47] | Wearable communication under 5G networks. | Presenting a multilayer communication architecture combines cloud/edge technologies, etc. | Efficiency, power efficiency, and connectivity can be enhanced by employing transmission technologies. |
| [36] | 5G wearable networks with cognitive computing. | Presenting the network slice-based 5G wearable networks and a data-driven resource management framework. | Improving the utilization ratio of resources. |
| [48] | Non-radiative wireless energy transfer. | Presenting the printed spiral resonator. | Power transfer efficiency = 74.96%. |
| [93] | A rectenna design at 28 GHz. | Creating a detector working at low input power levels. | Input power of 500 μW and responsivity of 96 V/W. |
| [49] | Characterizing the hand palm local channel. | Developing ultra-low power wireless sensors for wearable devices. | Path loss variation= higher than 3 dB. |

**Table 1.** *Cont.*

| Ref. | Scope | Contribution | Specifications |
|---|---|---|---|
| [50] | Wearable antennas for 5G systems. | Design of a dual-band dual-polarized button antenna. | Covering the 4.50–4.61 GHz band and the 5.04–5.50 GHz bands. |
| [51] | Sleep monitoring with low-cost WiFi devices. | Providing a fine-grained sleep log. | Demonstrating the effectiveness of SleepGuardian. |
| [52] | Headband for usage in human–machine interfaces and sleep data acquisition. | Using biomonitoring with wearable sensors. | Comfortable and low weight (115 g) to be used while sleeping. |
| [38] | Telemonitoring systems for 5G applications. | Used for patients with chronic respiration problems through 5G connectivity. | Successfully tested for 5G connectivity for at least 48 h. |
| [53] | Wearable multi-port antenna. | Monitoring of the human body by sampling health parameters. | Provided gain of −13 dBi. |
| [70] | Implant devices with electromagnetic coupling. | Proposing algorithm for coil couple geometry. | Power ranges as 10 W, 100 mW, and 300 µW. |
| [81] | Health monitoring and teleophthalmology. | Presenting template matching method for remote retina health monitoring. | Providing long-term retinal monitoring. |
| [82] | Gait phase recognition. | Presenting template matching method for remote retina health monitoring. | Short delay response less than 20 ms. |
| [76] | Biomedical image segmentation based on the neural network. | Presenting an evolutionary compression method. | Discovering efficient network architectures. |
| [83] | Cardiovascular implantable electronic devices. | Heart modeling with integrated stochastic optimization algorithm. | Practical physiological model to be used in clinical settings. |
| [65] | Point-of-care diagnostics with wearable devices. | A wireless 1.8-gram self-contained architecture. | Power consumption of 110 µW with latency of 10.2 µs. |
| [77] | Brain–machine interfaces with neural encoders. | Reproduces heterogeneous neural activity. | Enhances the fidelity of brain–machine interface simulators. |
| [71] | Biomedical electronics with segmented coils. | Designs small-scale wireless power transfer systems with optimizations. | Wireless power link at 40.68 MHz. |
| [78] | Designs a low-power device authentication system. | Designs ECG processor. | Improving area and power consumption by 64.5% and 51.0%. |
| [84] | Optimization for adaptive neuro-fuzzy inference system. | Identifies the heart condition by classifying the received sensor data. | Prediction model accuracy of 99.45%. |
| [85] | Custom foot orthoses for satisfactory treatment. | Presents the rapid evaluate and adjust device prescription methodology. | Improving the interaction between patient and physician. |
| [54] | Design of flow-guided nano-networks with optimizations. | Help in monitoring, data gathering, and transmission inside the human body. | Enhancing the network throughput of flow-guided nano-networks by selecting a suitable frame size. |

### 5.2. Verifying the Successful Collaboration

As presented in Figure 1, various data can be collected from patients using biomedical and implanted devices. Afterwards, these collected data through the wireless networks were transferred to the doctors located in the hospitals. As it is clear from both the transmitting and receiving sections, high-performance systems and devices must be installed for ensuring the successful proficiency among the various components and using 5G wireless networks.

### 5.3. Safety

Most medical devices have limited communication capabilities, and security has become a major concern in all devices. Hence, from the initial design step up to the final design step of medical devices, point-by-point factors such as medical tissue surroundings

and outcome specifications must be considered important. These considerations will result in more reliable and safety devices.

*5.4. Speed*

Due to the COVID-19 pandemic, the medical device industry has been affected to the highest degree. More than four million people have already been infected by this virus and the physical distance between the patient and doctor is significantly important. Hence, biomedical devices with the appropriate wireless networks must be arranged appropriately for transferring data. As described before, the 5G technologies are supporting high-data transportation and have suitable speed. For this case, it is recommended that this technology or even the sixth generation for future medical systems.

## 6. Conclusions

Biomedical devices play an important role in the healthcare communication systems. For effectively transferring data between the patient and medical doctors, emerging 5G and next generation technologies are required. Over the years, various medical and wearable devices have been presented that are more suitable for 5G technologies. The effectiveness of each device mainly depends on the applications to be used. This survey presents a comprehensive study on the introduction of various medical devices which may be used in next-generation 6G technology. It is well known that by increasing the technology in communication devices, advanced optimization methods are required for designing high-performance devices. From another point of view, this manuscript collects recently employed optimization methods for biomedical devices. Hence, researchers can easily determine the suitable wearable device for their communication technology and can select the optimization method that best fits their problem.

**Author Contributions:** Conceptualization, L.M., L.K., and I.P.; methodology, L.K.; software, L.K.; validation, L.K., L.M. and I.P.; formal analysis, L.M.; investigation, L.K. and I.P.; resources, L.M. and I.P.; data curation, L.K.; writing—original draft preparation, L.K.; writing—review and editing, L.K., L.M. and I.P.; visualization, L.K.; supervision, I.P. and L.M.; project administration, L.M. and I.P.; funding acquisition, I.P. All authors have read and agreed to the published version of the manuscript.

**Funding:** This work was partially supported by a grant of the Romanian Ministry of Education and Research, CNCS—UEFISCDI, project number PN-III-P4-ID-PCE-2020-0404, within PNCDI III.

**Institutional Review Board Statement:** Not applicable.

**Informed Consent Statement:** Not applicable.

**Data Availability Statement:** Not applicable.

**Conflicts of Interest:** The authors have no conflict of interest.

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
