# Peer review of "Magic of 5G Technology and Optimization Methods Applied to Biomedical Devices: A Survey"

_applsci, doi:10.3390/app12147096_

Round 1

Reviewer 1 Report

Authors have well addressed my concerns in the revised manuscript.

Reviewer 2 Report

In this resubmitted manuscript, the authors seem to address most of the comments of the reviewer. However, no point-to-point response is provided. It is rather difficult to characterize the revised content.

few more concerns for this revised manuscript. 

1. Why has the session 5-biomechanical devices from security issues? It talks about security challenges, why not put in session 6?

2. what are the differences between biomedical devices and wearable medical devices? Why the summary of the section in Figure 2 is totally different from the manuscript sections? 

3. The title, applied should be capitalized as Applied.

Reviewer 3 Report

The revised version still has numerous grammatical errors, and many expressions should be improved. Otherwise, it is satisfactory in the technical scope.

Minor Comment:

·      Fix numerous grammatical errors.

·      Remove Figure 2.
